# A Metalloproteinase Cocktail from the Venom of *Protobothrops flavoviridis* Cleaves Amyloid Beta Peptides at the α-Cleavage Site

**DOI:** 10.3390/toxins15080500

**Published:** 2023-08-12

**Authors:** Eugene Futai, Hajime Kawasaki, Shinichi Sato, Khadija Daoudi, Masafumi Hidaka, Taisuke Tomita, Tomohisa Ogawa

**Affiliations:** 1Laboratory of Enzymology, Graduate School of Agricultural Sciences, Tohoku University, Sendai 980-8572, Japan; hajime.kawasaki.r3@dc.tohoku.ac.jp (H.K.); daoudi.khadija.b8@tohoku.ac.jp (K.D.); masafumi.hidaka.a4@tohoku.ac.jp (M.H.); tomohisa.ogawa.c3@tohoku.ac.jp (T.O.); 2Frontier Research Institute for Interdisciplinary Sciences, Tohoku University, Sendai 980-8578, Japan; shinichi.sato.e3@tohoku.ac.jp; 3Laboratory of Neuropathology and Neuroscience, Graduate School of Pharmaceutical Sciences, The University of Tokyo, Bunkyo-ku, Tokyo 113-0033, Japan; taisuke@mol.f.u-tokyo.ac.jp

**Keywords:** α-secretase, a disintegrin and metalloproteinase (ADAM), Alzheimer’s disease, amyloid beta (Aβ), amyloid precursor protein (APP), snake venom metalloproteinase (SVMP)

## Abstract

A disintegrin and metalloproteinase (ADAM) family proteins are a major class of membrane-anchored multidomain proteinases that are responsible for the shedding of cell surface protein ectodomains, including amyloid precursor protein (APP). Human ADAM 9, 10, and 17 proteolyze APPs and produce non-amyloid-genic p3 peptides, instead of neurotoxic amyloid-β peptides (Aβs; Aβ40 and Aβ42), which form fibrils and accumulate in the brain of patients with Alzheimer’s disease (AD). The ADAM family is closely related to snake venom metalloproteinases (SVMPs), which are derived from ancestral ADAMs but act as soluble proteinases. To test the therapeutic potential of SVMPs, we purified SVMPs from *Protobothrops flavoviridis* venom using metal ion affinity and pooled into a cocktail. Thus, 9 out of 11 SVMPs in the *P. flavoviridis* genome were identified in the cocktail. SVMPs inhibited Aβ secretion when added to human cell culture medium without affecting APP proteolysis. SVMPs degraded synthetic Aβ40 and Aβ42 peptides at the same cleavage site (α-site of APP) as ADAM9, 10, and 17. SVMPs did not degrade Aβ fibrils but interfered with their formation, assessed using thioflavin-T. Thus, SVMPs have therapeutic potential for AD as an Aβ-degrading protease, and the finding adds to the discovery of bioactive peptides from venoms as novel therapeutics.

## 1. Introduction

A disintegrin and metalloproteinase (ADAM) family proteins are a major class of membrane-anchored multidomain proteinases that are responsible for the ectodomain shedding of cell surface proteins, including growth factors, cytokines and their receptors, and cell adhesion molecules [1]. ADAMs are type-I transmembrane proteins harboring metalloproteinase, disintegrin-like, cysteine-rich, epidermal growth factor (EGF), and transmembrane domains (Figure 1a). To date, 20 ADAM family genes have been identified in the mammalian genome [1]. However, only 12 human ADAMs (ADAM 8, 9, 10, 12, 15, 17, 19, 20, 21, 28, 30, and 33) contain a functional catalytic consensus sequence; the others (ADAM 2, 7, 11, 18, 22, 23, 29, 32) are proteinase-inactive pseudoenzymes of unclear physiological function. Some ADAMs (ADAM 9, 12, and 28) have splicing variants lacking transmembrane and cytoplasmic regions [1], and they are expressed as soluble glycoproteins. There exist two closely related ADAM proteins: ADAM with thrombospondin motifs (ADAMTS) and snake venom metalloproteinase (SVMP). 

The ADAMTS family is a close relative of the ADAM family and is important in various physiological processes, including extracellular matrix remodeling and blood coagulation [1,2]. ADAMTS proteins contain varying numbers of thrombospondin motifs and different domains in place of the transmembrane and cytoplasmic domains (Figure 1a) and function as secreted proteinases. The 19 ADAMTS proteins can be sub-grouped according to their substrates as proteoglycanases (ADAMTS1, 4, 5, 8, 9, 15, and 20), procollagen N-propeptidases (ADAMTS2, 3, and 14), cartilage oligomeric matrix protein cleaving proteinases (ADAMTS7 and 12), von Willebrand factor (VWF) cleaving proteinase (ADAMTS13), and a group of orphan proteinases with unknown substrates (ADAMTS6, 10, 16, 17, 18, and 20).

SVMPs are major components of most viper venoms and function as soluble proteinases [1]. Proteomic analyses revealed that SVMPs constitute more than 30% of the total protein in many Viperidae venoms, including that of *Protobothrops flavoviridis* [3]. SVMPs have hemorrhagic activity and may also interfere with the hemostatic system in animals. SVMPs are categorized into three classes (P-I, P-II, and P-III) according to their domain organization (Figure 1a). P-III SVMPs have a modular structure homologous to the ectodomain of ADAMs, similar to soluble ADAM variants. SVMPs exhibit fibrinogenolytic or fibrinolytic activity, prothrombin or factor X activation, and inhibition of platelet aggregation, which are linked to the functions of ADAMTS proteins [2]. P-III SVMPs have higher hemorrhagic activity than P-I SVMPs because their large carboxyl terminus domains endow resistance to inhibition by the plasma proteinase inhibitor α2-macroglobulin [4]. *Protobothrops flavoviridis* has 11 SVMPs [5,6], which were generated through recruitment, duplication, and neofunctionalization of an ancestral gene; they are most closely related to ADAM7, 28, and ADAMDEC-1 (decysin-1), a unique protein with solely the metalloproteinase domain [1,7]. Thus, SVMPs are proteins of the ADAM family, and they are also referred to as snake ADAMs. 

Alzheimer’s disease (AD) is a neurodegenerative disease characterized by the accumulation of amyloid-beta (Aβ) plaques in the brain [8]. Two shedding proteases, beta-site amyloid precursor protein cleaving enzyme-1 (BACE-1) and ADAMs, are responsible for the ectodomain shedding of amyloid precursor protein (APP) and have been implicated in the pathogenesis of AD [9] (Figure 1b). The sequential cleavage of APP by BACE-1 and γ-secretase produces amyloidogenic Aβ peptides of different lengths: Aβ38, Aβ40, Aβ42, and Aβ43 (composed of 38–43 amino acids). By contrast, the cleavage of APP by ADAMs and γ-secretase yields non-amyloidogenic p3 peptides. In AD, abnormal cleavage of APP by BACE-1 or γ-secretase leads to the production of neurotoxic Aβs, i.e., Aβ42 and Aβ43 [10,11]. Inhibiting BACE-1 and γ-secretase activity has been explored as a potential therapeutic strategy to reduce neurotoxic Aβ production and slow disease progression [9,12,13]. Activating ADAM protease through the signal transduction pathways involved in the regulation of ADAM activity and increasing non-amyloidogenic processing also have therapeutic potential [9,14]. The activators of ADAM, BACE1 inhibitors, γ-secretase inhibitors, and γ-secretase modulators have been developed and evaluated [9,12,13,14]; however, all have failed in clinical trials as far. The currently approved therapy for AD includes cholinesterase inhibitors, N-methyl-D-aspartate (NMDA)-receptor antagonists, and their combination therapy, but these provide only temporary symptomatic relief [15]. Recently, anti-Aβ antibodies have been shown to mediate the removal of amyloid plaque from the brains of patients with AD, and the U.S. Food and Drug Administration (FDA) has granted accelerated approval to two of these, aducanumab and lecanemab [16,17]. While these drugs offer hope for a cure for AD, they also present problems, including individual differences in efficacy, side effects, and high treatment costs. Novel therapeutic strategies are still required.

Although snake venom is typically cytotoxic, neurotoxic, and/or hemotoxic, its medical potential has long been recognized [3,18]. Hemotoxins have been developed into approved drugs, including the antihypertensive captopril [19], the antiplatelets tirofiban [20] and eptifibatide [21], the drug batroxobin for acute cerebral infarction [22], the anti-hemorrhagic haemocoagulase [23], and the analgesic α-cobrotoxin [23]. Among these, captopril, tirofibran, eptifibatide, and α-cobrotoxin are small-peptide drugs, whereas batroxobin and haemocoagulase are proteases. For example, batroxobin induces the degradation of fibrinogen [22]. We hypothesized that SVMPs may have therapeutic potential for AD targeting Aβ.

In this study, we evaluated the function of SVMPs in APP processing. We purified *P. flavoviridis* SVMPs and incorporated them into a human cell culture medium at a concentration lower than the cytotoxic concentration to evaluate their therapeutic potential for AD by assessing their effects on Aβ secretion and APP levels. We found that SVMPs cleave Aβ at the α-cleavage site of APP without affecting APP processing. We further examined whether SVMPs were capable of degrading Aβ fibrils.

## 2. Results

### 2.1. Fractionation of Snake Venom Metalloproteinases

We fractionated the crude venom of *P. flavoviridis* via gel filtration chromatography to high- (>50 kDa), medium- (20–50 kDa), and low- (<20 kDa) molecular-weight fractions (Figure 2a). The medium fraction encompassed the major protein peak and contained P-I, P-II, and P-III SVMPs. To fractionate SVMPs, we subjected the medium fraction to metal chelation affinity chromatography purification, according to a report on the purification of SVMPs from venom of the viper *Trimeresurus albolabris* [24]. SVMPs were eluted with imidazole and prepared as a cocktail after desalting and concentration through ultrafiltration. Sodium dodecyl–sulfate polyacrylamide gel electrophoresis (SDS–PAGE) of the SVMP fraction yielded multiple bands corresponding to P-I and P-III SVMPs (Figure 2b,c). Further, 1.7 mg of SVMPs was purified from 20.3 mg of crude venom (Appendix A).

SVMP activity was evaluated via a casein degradation assay (Figure 2d). Casein-degrading activity was detected in crude venom, the gel filtration (GF) medium-molecular-weight fraction, and the affinity-purified (AF) fraction. The activity of each fraction was greatly reduced in the presence of EDTA, an SVMP inhibitor, indicating that protease activity in the fractions was mediated by SVMPs. Casein-degrading activity was highest in the AF fraction (Figure 2d). The specific activity of the AF fraction increased by approximately 1.5-fold compared to the GF fraction and 2-fold compared to crude venom. The recovery of SVMP activity in the AF fraction was 22.8% of the GF fraction and 16.7% of the crude venom (Appendix A).

In the AF fraction, liquid chromatography–tandem mass spectrometry (LC–MS/MS) identified 9 of the 11 SVMPs [5,6] discovered in the *P. flavoviridis* genome (Table 1). The sequences of these SVMPs are provided in Appendix A. The purified SVMP cocktail contained SVMPs of classes P-I, P-II, and P-III. Two SVMPs, NaMP and elegantin, were not identified using LC–MS/MS, despite the presence of their encoding genes. They were also not detected in prior proteomic studies of *P. flavoviridis* crude venom (unpublished data), possibly due to their low expression levels.

### 2.2. SVMPs Reduced Aβ Production by Cultured Human Cells

SVMP fractions were subjected to a cell-based assay of Aβ production. We analyzed the toxicity of the SVMP fractions to human embryonic kidney (HEK) 293 cells stably expressing APP Swedish mutant (NLNTK). Cell viability was assessed by measuring dehydrogenase activity using a Cell Counting Kit-8 (Dojin Chemical Co., Ltd., Kumamoto, Japan). The cell viability rate was ~100% with 2, 4, and 8 μg/mL SVMPs but decreased to ~80% with 16 μg/mL SVMPs and to <40% with 32 μg/mL SVMPs (Figure 3a). IC_50_ values were 24.8 μg/mL SVMPs with 95% confidence intervals from 20.6 to 29.9 μg/mL SVMPs. Phase-contrast microscopy showed many floating cells with 16 and 32 μg/mL SVMPs but very few with 2–8 μg/mL SVMPs. Based on these results, we used SVMPs at 5 μg/mL in subsequent experiments. 

NLNTK cells were cultured in a medium supplemented with 5 μg/mL SVMPs for 48 h, and secreted Aβ and intracellular APP fragments (C99 or C83) were analyzed via immunoblotting. SVMPs reduced the level of Aβ to 10% of the control (no SVMPs) (Figure 3b,c). By contrast, the levels of APP carboxy-terminus fragments (C99 and C83) were unaffected by SVMP (Figure 3b,d,e). These results suggest that SVMPs can degrade secretory Aβ peptides without affecting APP proteolysis.

### 2.3. SVMPs Cleaved Aβ40 and Aβ42 In Vitro at α-Cleavage Sites 

Because SVMPs act directly on Aβ, we next analyzed their ability to degrade synthetic Aβ peptides in vitro. Immunoblotting showed that SVMPs reduced the levels of Aβ40 and Aβ42 compared to the control (no SVMPs) (Figure 4a, lanes 1–3 and 7–9, Figure 4b). In addition, the degradative activity of SVMPs was markedly reduced by EDTA, an SVMP inhibitor. These results indicate that SVMPs can degrade Aβ40 and Aβ42 in vitro. 

To identify the SVMP cleavage site in Aβ, we performed LC–MS/MS (Table 2 and Table 3). The amino-terminal fragments of the Aβ peptide were identified. Based on the peptide fragments detected through LC–MS/MS, the cleavage sites of Aβ40 were Aβ Lys_16_-Leu_17_, Val_18_-Phe_19_, and Ala_30_-Ile_31_. The Lys_16_-Leu_17_ cleavage site in Aβ corresponds to APP Lys_687_-Leu_688_, which is the α-cleavage site at which APP is degraded by ADAM9, 10, and 17. The SVMPs cleaved Aβ at the same cleavage site as ADAMs. Therefore, we term the Aβ Lys_16_-Leu_17_ cleavage site the α-cleavage site of Aβ. The highest intensity was detected via LC–MS/MS in the amino terminal peptides cleaved at the α-site (99.5% of Aβ40-degraded peptides and 80.6% of Aβ42-degraded peptides), suggesting that the α-cleavage site is the principal cleavage site. 

### 2.4. SVMPs Did Not Cleave Thioflavin-Positive Aβ Fibers but Inhibited Their Formation

Aβ fibrils were formed in Tris-buffered saline based on previous studies, and amyloids were detected by measuring the fluorescence intensity of Thioflavin T (ThT). Initially, we investigated whether SVMPs degrade Aβ40 or Aβ42 fibrils. SVMPs were added to the solution after Aβ fibril formation, incubated for 24 h, and ThT was added. The ThT fluorescence intensity was unaffected by SVMPs (Figure 5a), suggesting that SVMPs did not degrade amyloid-structured Aβ40 or Aβ42. Then, we investigated whether SVMPs can inhibit the amyloid formation of Aβ. SVMPs were added to Aβ fibrillation solution and incubated for 24 h. In the control, ThT fluorescence increased ~5.4-fold during the experiment, and SVMPs significantly (2.3-fold) suppressed the increase in ThT fluorescence (Figure 5b). By contrast, in the presence of EDTA, ThT fluorescence increased 3.3-fold, indicating reduced suppression by SVMPs. These results suggest that the Aβ40-degrading activity of SVMP inhibits amyloid fibril formation.

## 3. Discussion

As the number of patients with AD increases in aging populations, the development of novel therapies increases in importance [17]. In this study, we aimed to develop therapeutics for AD based on amyloid cascade theory. SVMPs are related to ADAM proteins, which cleave APP at the α-site and have α-secretase activities. SVMPs differ from ADAMs because SVMPs lack transmembrane domains and act as soluble proteinases. In this study, unlike ADAMs, *P. flavoviridis* SVMP efficiently cleaved Aβ but not APP. Moreover, SVMP did not cleave aggregated amyloid fibers but decreased the number of amyloid fibers during Aβ fiber formation.

Although many drugs that target Aβ in AD have failed to demonstrate clinical efficacy, four anti-Aβ antibodies have been shown to mediate the removal of amyloid plaque from the brains of patients with AD. The FDA has granted accelerated approval to two of these, aducanumab and lecanemab [16,17]. Aβ molecules aggregate to form oligomers and protofibrils (PFs), which are deposited as insoluble amyloid plaques. Aducanumab recognizes amino acids 3–7 of Aβ and is specific to amyloid plaques. Lecanemab recognizes soluble protofibrils of Aβ. Among the four anti-Aβ antibodies, lecanemab bound most strongly to Aβ protofibrils and showed the greatest Aβ removal, whereas the others preferred more highly aggregated forms [17]. By contrast, anti-Aβ antibodies that predominantly bind to monomeric Aβ do not mediate plaque removal and lack clinical efficacy [17]. Success in immunotherapy-targeting inhibition of Aβ aggregation open a new route to other therapy targeting Aβ. The current anti-Aβ antibodies have problems in low efficacy, side effects, and high treatment costs. Based on their proven ability to remove Aβ, Aβ-degrading proteases including SVMPs have therapeutic potential.

A number of human Aβ-degrading proteases have been identified. These comprise proteases with different catalytic sites. For example, there are 13 metalloproteases, including neprilysin (NEP), insulin-degrading enzyme (IDE), endothelin-converting enzyme (ECE-1, -2), and matrix metalloproteinases (MMP-2 -3, -6, -9 and -14); 4 serine proteases, including kallikrein-related peptidase 7 (KLK7), plasmin and myeline basic protein; 3 aspartic proteases, cathepsin D, BACE1, and BACE2; 1 cysteine protease, cathepsin B; and 1 threonine protease, proteasome [25,26,27]. SVMPs from snake venom are added to the list of Aβ-degrading metalloproteases. These proteases show different enzymatic characteristics and a distinct effect on the Aβ monomer, oligomer, and fibril. Like SVMPs, NEP and IDE degrade monomeric Aβ but not Aβ fibrils. On the other hand, MMPs and KLK7 degrade both Aβ monomers and Aβ fibrils [26]. Clearance of Aβ was decreased in patients with sporadic AD, comprising ~95% of all patients with AD. Studies of human Aβ-degrading proteases have provided insight into the pathogenesis of AD [25]. 

To promote the development of Aβ-degrading protease-based therapeutics, it is essential to establish methods to enhance or recover protease activity. Using a recombinant adeno-associated virus system to overexpress a gene of interest in neurons administered in peripheral blood, NEP was overexpressed in neurons throughout the brain [28]. NEP reduced the brain Aβ level and the amyloid burden in a mouse model of AD. In addition, diazoxide-mediated enhancement of NEP via regulation of the somatostatin pathway was effective in a mouse AD model [29]. Simple overexpression of Aβ-degrading proteases may not be enough to reduce the Aβ burden. For MMPs, tissue inhibitors of MMP (TIMPs) are endogenous inhibitors. The expression of TIMP-1 was reported to increase in the cerebrospinal fluid (CSF) of AD patients [30]. Administration of exogenous SVMPs may have advantages with low-endogenous inhibitors. To reduce the brain Aβ level and amyloid burden, SVMPs could be administered intravenously or intrathecally; however, there is difficulty crossing the blood–brain barrier (BBB). Ziconotide, a ω-conotoxin peptide from magic cone snail (*Conus magus*) venom, is a neurotoxin-derived analgesic for severe chronic pain [31]. Although the efficacy of ziconotide is 100–1000-fold that of morphine, it had difficulty in crossing the BBB and is administered intrathecally. This is a common problem facing venom-derived neuropeptide drugs. The possibility of using AAVs should be considered in the future.

In this study, SVMP showed high proteolytic activity against monomers but not fibrils, which is detectable via ThT fluorescence. We incubated the reaction for 24 h, which is a shorter period than in previous studies. MMP9 reduced the ThT signal significantly after 5 days [30] and myeline basic protein after 2 days [32]. SVMPs may reduce the ThT signal over the long term. Importantly, SVMPs cleave soluble Aβ more efficiently than Aβ fibrils. Indeed, NEP reportedly cannot degrade Aβ fibrils [30] but showed clinical efficacy when introduced into the brain. Therefore, we believe that SVMP has therapeutic potential.

Two-step purification through gel filtration and affinity purification was performed in this study, as for *T. albolabris* SVMPs [24]. SVMPs interacted with the NTA resin via the catalytic Zn^2+^ ion and were dissociated by 10–30 mM imidazole. The AF fraction had high casein-degrading activity, which was ~1.5-fold that of the GF fraction. These results indicate no significant loss of activity and that SVMPs in the imidazole elution fraction retained their catalytic Zn^2+^ during affinity purification. In an experiment using cultured cells, a cocktail of SVMPs was added to the medium. Robust Aβ degradation was observed at sufficiently low concentrations (5 μg/mL) compared to the IC_50_ (24.8 μg/mL) for toxicity. Further, 90% of secreted Aβ was degraded. Considering why the 10% Aβ remains in the medium, it is possible that the rate of Aβ degradation balances the rate of Aβ secretion, or it could be due to the generation of Aβ fibril that SVMPs do not degrade. These possibilities should be clarified in future studies. The purified fraction containing nine SVMPs was used as a cocktail, precluding identification of the active SVMP, which should be the focus of a future study. The nine SVMPs include flavoridin, flavorase, and jerdonitin-like, which inhibit platelet aggregation; HR1a, HR1b, and HR2a, which exhibit bleeding activity; VMP III-like, which exhibits fibrinolytic activity; HV1, which induces apoptosis; and H2pro with unknown function [5,6]. To develop therapeutics, it is necessary to balance cytotoxicity and Aβ-degrading activity. It is necessary to select SVMPs with high Aβ-degrading activity and low toxicity. Finally, the functions of SVMPs need to be verified in vivo using a mouse model of AD. 

We identified three SVMP-cleavage sites in Aβ40 and Aβ42; the major cleavage site was the α-site Aβ Lys_16_-Leu_17_, which is the same site targeted by α-secretase (ADAM9, 10, 17) to cleave APP. SVMPs cleave APP monomers with higher efficiency than fibrils, suggesting that the substrate recognition pockets of SVMPs are similar to those of ADAM9, 10, and 17. The other cleavage sites in Aβ are close to the α-site and in the transmembrane region of APP Val_18_-Phe_19_ and Ala_30_-Ile_32_. All cleavage sites are the amino termini of hydrophobic residues, suggesting that these SVMPs possess hydrophobic P1′ pockets, a characteristic of ADAM/SVMP proteins [33]. Although the mechanism of APP recognition by ADAM is unknown, analysis of the interaction between SVMP and Aβ may provide insight into this matter. 

Low-density lipoprotein receptor-related proteins 5 and 6 (LRP5/6), which are Wnt receptors, are cleaved by *Crotalus atrox* SVMP (vascular apoptosis-inducing protein 1, VAP-1) or ADAM8/12 at the same site [34] as for SVMPs and ADAMs. ADAM9s, which lack a transmembrane domain, have a similar structure to SVMPs. Co-expression of ADAM9s with APP in cultured cells increased sAPPα production from APP [35]. Although the role of ADAM9s in Aβ clearance remains unclear, if it exhibits Aβ-degrading activity, ADAM9s could be a target for the development of novel drugs for AD. These results indicate that the *Crotalus atrox* SVMP or soluble ADAM variant shares transmembrane substrates with ADAMs. In this report, we found that *P. flavoviridis* SVMPs did not cleave APP. Future analyses should clarify why *P. flavoviridis* SVMP could not cleave APP on the membrane.

## 4. Materials and Methods

### 4.1. Materials

*Protobothrops flavoviridis* (Amami-Ohshima) crude venom was provided by the Kagoshima Prefectural Public Health Office in Japan. The crude venom was a mixture of unknown origin, either male or female snakes, as it was collected without regard to sex. The NLNTK cells are the human embryonic kidney (HEK) 293 cell (CRL-1573 in ATCC, Manassas, VA, USA) line stably expressing APP Swedish mutant, prepared as previously [36]. Synthetic amyloid beta peptides Aβ40 and Aβ42 were purchased from Peptide Institute, Inc. (Osaka, Japan). Monoclonal antibodies against Aβ N-terminus (82E1) and tubulin (DM1A) were obtained from Immuno-Biological Laboratories (Fujioka, Japan) and Novus Biologicals (Littleton, CO, USA), respectively. Polyclonal antibodies against the carboxy terminal of APP (A8717), and thioflavin T (2-[4-(dimethylamino)phenyl]-3,6-dimethyl-1,3-benzothiazol-3-ium chloride, ThT) were obtained from Sigma-Aldrich (St. Louis, MO, USA). All other reagents were of commercial grade and were purchased from Wako Pure Chemicals (Osaka, Japan) or Nacalai Tesque (Kyoto, Japan).

### 4.2. Fractionation of SVMPs from Crude Venom of P. flavoviridis

Crude venom of P. flavoviridis was separated via gel filtration, as described previously [37]. Lyophilized venom (3.0 g) was dissolved in 50 mM CH_3_COONH_3_ in 0.1 mM CaCl_2_, applied to a Sephadex G-100 column (ø2.8 × 75 cm, Cytiva, Marlborough, MA, USA) and separated using 50 mM CH_3_COONH_3_ in 0.1 mM CaCl_2_ at a flow rate of 0.32 mL/min. Fractions (7 mL) were collected and pooled as high- (>50 kDa, G1), medium- (20–50 kDa, G2) and low- (<20 kDa, G3) molecular-weight fractions. Each fraction was lyophilized and stored at −80 °C until use. To purify SVMPs, the medium-molecular-weight G2 fraction was subjected to metal chelation affinity chromatography, as described for the purification of *T. albolabris* SVMPs [24]. The lyophilized G2 fraction (1.0 mg) was reconstituted in 1.0 mL of phosphate-buffered saline, and the insoluble fraction was removed through centrifugation at 20,600× *g* for 2 min. Soluble supernatant was mixed with 1.0 mL of pre-washed His 60 Ni Superflow Resin (TaKakaRa Bio Inc., Shiga, Japan) in a polypropylene column (Muromac mini; Muromachi Chemical, Fukuoka, Japan) and incubated overnight at 4 °C with inversion to bind the SVMPs to the resin. Next, unbound fractions were collected as the flowthrough (FT). The column was washed with 10 mL of wash buffer (10 mM Tris/HCl (pH 8.0), 150 mM NaCl), and 10 wash fractions (W1–10, 1 mL each) were collected. Elution buffer containing imidazole (10 mM Tris/HCl (pH 8.0), 150 mM NaCl, 10–30 mM imidazole) was applied sequentially to the column; 2 mL of 10 mM imidazole, 2 mL of 20 mM imidazole, 2 mL of 30 mM imidazole, and 1 mL of step-gradient eluted fractions were collected (E1–E6). The SVMP-containing fractions, E3–E5, were pooled and ultrafiltered to exchange buffer with phosphate-buffered saline and concentrate proteins using an Amicon Ultra-15 (Millipore, Burlington, MA, USA, MWCO 10,000). The purified SVMP fractions were frozen in liquid nitrogen and stored at −80 °C. The SVMP fractions were confirmed using SDS–PAGE, and the protein concentration was determined via Bradford Protein assay (Bio-Rad, Hercules, CA, USA). 

### 4.3. Caseinolytic Assay

A caseinolytic assay was performed [38] to evaluate SVMP activity. Briefly, 30 mg of casein was added to 1 mL of 1 M Tris-HCl (pH 8.0) and mixed by inversion at 4 °C overnight. The solution was heated at 40 °C for 10 min to dissolve casein completely. The casein solution was diluted to create assay solutions (total volume: 250 μL, 3 mg/mL casein, 10 μg/mL enzyme solution, 100 mM Tris-HCl (pH 8.0) with or without 5 mM EDTA). The reaction was started with the addition of 25 μL enzyme solution, i.e., the crude venom, G2, or AF fractions (100 μg/mL protein), and incubated at 37 °C for 2.5 h. The reaction was stopped by adding 250 μL of 10% TCA, incubated on ice for 20 min, and centrifuged at 20,600× *g* for 10 min. The A_280_ of the supernatant was measured; an increase in A280 of 1.0 in 1 min was defined as 1 unit of caseinolytic activity.

### 4.4. Cytotoxicity Assay

SVMP cytotoxicity was measured using the Cell Counting Kit-8 (Doujin Chemical Co., Ltd., Kumamoto, Japan). Briefly, 4.0 × 10^4^ NLNTK cells [36] were cultured in 96-well plates with Dulbecco’s modified Eagle’s medium (DMEM) supplemented with 10% fetal bovine serum (FBS) and 1% penicillin and streptomycin at 37 °C and 5% CO_2_ in a tissue culture incubator. After 24 h, the medium was changed to DMEM without FBS or antibiotics containing SVMP (2, 4, 8, 16, 32 μg/mL) and incubated for 48 h. Next, the medium was changed to 100 μL of fresh DMEM with no serum, and 10 μL of Cell Counting Kit-8 reagent was added to each well and incubated for 30 min. Next, the A_450_ was measured using an iMark microplate reader (Bio-Rad). Cell viability (%) was calculated by considering a sample without SVMP as showing 100% cell viability. The IC_50_ value of SVMP was determined by fitting the survival curve to the equation: Y = Y_o_/(1 + 10^((LogIC_50_−X) × HillSlope)), where Y is the cell viability (%), Y_o_ is the top value, and X is the Log[SVMP]. Curvefit was performed in GraphPad Prism 5 (GraphPad Software, Boston, MA, USA).

### 4.5. Aβ Production in NLNTK Cells

Aβ metabolism was analyzed in NLNTK cells [36]. Cells were cultured in DMEM with 10% FBS and 1% penicillin and streptomycin and maintained at 37 °C and 5% CO_2_ in a tissue culture incubator. The medium was replaced with DMEM containing 5 μg/mL SVMP with no serum or antibiotic. After incubation for 48 h, the conditioned medium and cells were recovered. The secreted Aβ level in the medium, the intracellular levels of APP fragments (APP C83 and C99), and the intercellular level of tubulin were measured through immunoblotting after conventional 16.5% polyacrylamide Tris/Tricine gel electrophoresis [39]. Tris/Tricine gel electrophoresis was suited to separate small peptides, like Aβ, C83, and C99. Blots were developed using an ECL system, and band intensities were quantified on an LAS-4000 Image Analyzer (Fujifilm, Tokyo, Japan); a control Aβ marker was used to assess the Aβ concentration.

### 4.6. Proteolysis of Aβ40 and Aβ42 Peptides

Synthetic Aβ40 and Aβ42 peptides were mixed with SVMP fractions as follows: 10 nM Aβ40 or Aβ42, 60 μg/mL SVMP, and 100 mM Tris-HCl (pH 8.0) with or without 5 mM EDTA. The reactions were incubated at 37 °C for 2.5 h, and Aβ levels were quantified via immunoblotting, as described in Section 4.5. To identify cleavage sites, Aβs were digested under the following conditions: 0.1 mg/mL Aβ40 or Aβ42, 5.6 μg/mL SVMP, and 100 mM Tris-HCl (pH 8.0). The mixtures were incubated at 37 °C for 24 h and subsequently subjected to LC–MS/MS.

### 4.7. Aβ Fibril Formation Detection through ThT Assay

Aβ fibrils were detected using a thioflavin T (2-[4-(dimethylamino) phenyl]-3,6-dimethyl-1,3-benzothiazol-3-ium chloride, ThT) (Sigma-Aldrich) assay, as described previously [40]. The ThT assay is a β-sheet-specific fluorescence assay. Briefly, synthetic Aβ40 and Aβ42 peptides were mixed as follows: 0.1 mg/mL Aβ40 or Aβ42, 100 mM Tris-HCl (pH 8.0), and 50 mM NaCl. The mixture was vortexed briefly and incubated at 37 °C for 24 h. At 0 and 24 h, 5 μL of fibril mixture was aliquoted into a 96-well plate, mixed with 200 μL ThT solution (5 μM ThT, 100 mM glycine (pH 7.5)), and incubated for 5 min at room temperature in the dark. ThT fluorescence was analyzed using a Gemini XPS microplate spectrofluorometer (Molecular Devices, San Jose, CA, USA) in endpoint mode with excitation and emission wavelengths of 440 and 480 nm, respectively. SoftMax Pro Software v5.4.1 (Molecular Devices) was used for data analysis. Fluorescence intensities were used after subtracting the value of the control (ThT solution without Aβ).

To analyze their ability to degrade Aβ fibrils, SVMPs were added before or after fibril formation, and the mixture was incubated for 24 h and subsequently subjected to a ThT assay using a control solution without SVMP.

### 4.8. LC-MS/MS Conditions

The affinity-purified SVMP fractions were digested with lysyl-endopeptidase, as described previously [41]. Briefly, 100 µg of SVMP fractions was dissolved in 8 M urea, 0.4 M ammonium hydrogen carbonate, and 10 mM DTT and incubated at 50 °C for 15 min. Iodoacetamide was added to a final concentration of 10 mM and incubated for 15 min at room temperature. The reaction was diluted 4-fold with water to reduce the urea concentration to 2 M. Next, lysyl-endopeptidase was added and incubated overnight at 37 °C. Next, samples were desalted using ZipTip C18 pipette tips (Millipore) according to the manufacturer’s instructions, desalted using C18 pipette tips (Nikkyo Technos Co., Ltd., Tokyo, Japan), and subjected to nano-LC–MS/MS. 

Nano-LC–MS/MS analysis was performed using an LC–nano-ESI–MS comprising a quadrupole time-of-flight mass spectrometer (Triple TOF 5600 system; SCIEX, Framingham, MA, USA) equipped with a nanospray ion source and a nano-LC system (Eksigent; SCIEX). Peptides (2 μL) were injected into a trap column (NanoLC Trap ChromXP C18, 3 μm, 120 Å, SCIEX) and an analytical column (Nano HPLC capillary column, 75 μm × 155 mm, 3 μm, ODS, Nikkyo Technos), followed by elution with buffer A (0.1% formic acid) and buffer B (100% acetonitrile and 0.1% formic acid). The peptides were eluted in a linear gradient of 0% to 40% buffer B for 20 min, 40% to 98% buffer B for 1 min, and 98% buffer B for 6 min at a flow rate of 300 nL/min. Nano-LC–MS/MS data were acquired in data-dependent acquisition mode using the Analyst TF 1.5.1 software (SCIEX). The data-dependent acquisition settings were as follows: accumulation time of 0.25 s; full MS scan range of 400–1250 *m/z*, excluding the former target ion for 12 s; mass tolerance of 50 mDa; and selection of the top 10 signals for MS2 scanning in each full MS scan. The experiment was conducted in triplicate. The MS/MS spectra were searched against the protein sequence databases derived from the RNA-seq data of *P. flavoviridis* snake venom using MaxQuant (freeware) with the default settings. The peptide sequences were matched against the National Center for Biotechnology Information (NCBI) non-redundant protein database of *P. flavoviridis* (taxid 88087) using BLASTP (http://blast.ncbi.nlm.nih.gov accessed on 19 September 2022) and amino acid sequences derived from transcriptomics data for *P. flavoviridis* venom glands [6].

Aβ fragments generated by SVMPs were subjected to LC–MS/MS. After the Aβ40 and Aβ42 peptides were digested with SVMPs, formic acid was added to a final concentration of 0.1%. The samples were desalted using ZipTip C18 pipette tips (Millipore) and C18 pipette tips (Nikkyo Technos Co., Ltd.) and subjected to nano-LC–MS/MS, as described above.

### 4.9. Statistical Analysis

Statistical comparisons were performed using the Student’s t-test in Figure 3c–e, the one-way analysis of variance followed by Dunnett’s multiple comparisons test in Figure 2d and 3a, or Turkey’s multiple comparisons test in Figure 4b and Figure 5, using GraphPad Prism 5 (GraphPad Software).

## Figures and Tables

**Figure 1 toxins-15-00500-f001:**
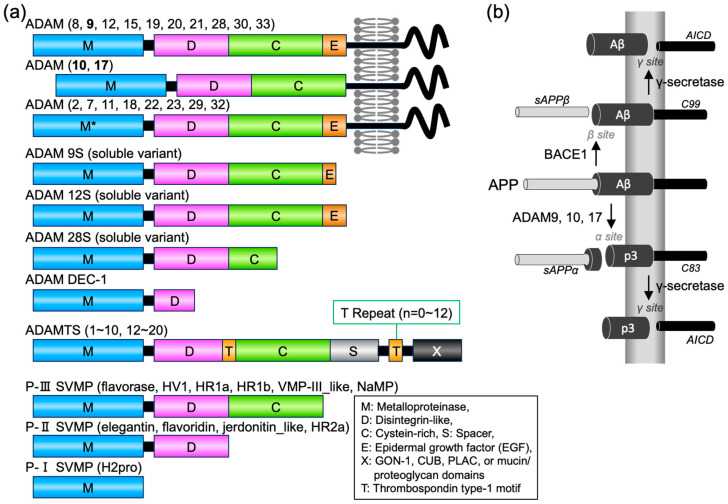
A disintegrin and metalloproteinase (ADAM) family proteases and amyloid precursor protein (APP) proteolytic pathways. (**a**) Schematic diagram of the domain structures of ADAM, ADAM with thrombospondin motifs (ADAMTS), and snake venom metalloproteinase (SVMP) family proteins. Colors indicate domains or motifs. M, D, C, E, and S indicate metalloproteinase, disintegrin-like, cysteine-rich, epidermal growth factor (EGF), and spacer domains, respectively. M* indicates metalloproteinase domain without protease activity. T, thrombospondin type-I motif. X, additional domains in ADAMTS proteins, including GON-1, CUB (complement components C1rC1s/urinary EGF/bone morphologenic protein-1), PLAC (protease and lacunin), and mucin/proteoglycan domains. For *Protobothrops flavoviridis* SVMPs, the names of six P-III genes, four P-II genes, and one PI gene are provided. (**b**) Human APP proteolysis via the non-amyloidogenic and amyloidogenic pathways. Non-amyloidogenic processing of APP refers to its sequential processing by ADAM9, 10, and 17 (domain structure are shown bold in (**a**)), which cleave the Aβ region to generate the membrane-tethered C-terminal fragment C83 and the N-terminal fragment sAPPα. C83 is next cleaved by γ-secretases to generate extracellular p3 peptide and the APP intracellular domain (AICD). Amyloidogenic processing of APP is carried out by the sequential action of membrane-bound BACE1 and γ-secretases. BACE1 cleaves APP into the membrane-tethered C-terminal fragment C99 and N-terminal fragment sAPPβ. C99 is subsequently cleaved by γ-secretases into the extracellular Aβ peptides and AICD.

**Figure 2 toxins-15-00500-f002:**
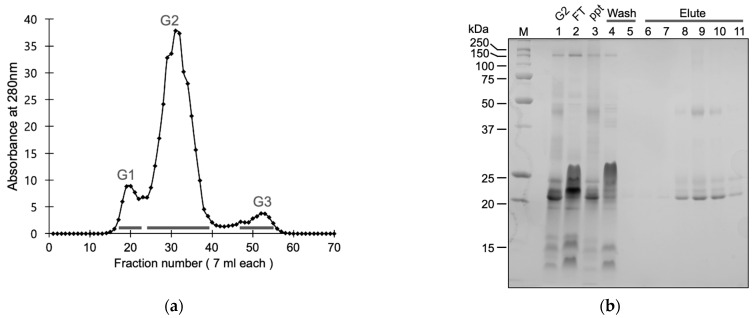
Purification and analysis of SVMPs from *P. flavoviridis* venom. (**a**) Gel filtration chromatography of the venom using a Sephadex G-100 column (ø2.8 × 75 cm) at flow rate of 0.32 mL/min with 50 mM CH_3_COONH_3_ in 0.1 mM CaCl_2_. Three fractions (G1, G2, and G3) corresponding to high- (>50 kDa), medium- (20–50 kDa), and low- (<20 kDa) molecular-weight fractions were collected. (**b**) The lyophilized G2 fraction was reconstituted and purified via nickel-affinity chromatography. G2, flowthrough FT, pellet ppt, Wash (W1 and W10) and imidazole-eluted elute fractions (E1–E6) were separated by non-reducing sodium dodecyl–sulfate polyacrylamide gel electrophoresis (SDS–PAGE) and stained with Coomassie brilliant blue. For non-reducing SDS-PAGE, samples were prepared in SDS sample buffer without thiol reagents and loaded on 12.5% polyacrylamide gel (size 10 cm × 10 cm) with standard running buffer (25 mM Tris/HCl [pH 8.3], 192 mM Glycine, 0.1% SDS). Ppt represents the insoluble fraction after reconstitution of G2. (**c**) The crude venom (CV), medium molecular weight (G2) gel filtration (GF), and affinity-purified (AF) fractions were separated by non-reducing SDS–PAGE and stained with Coomassie brilliant blue. Arrows indicate bands corresponding to P-I and P-III SVMPs after the removal of signals and prodomains. 20.3 mg of crude venom was used for purification. The recovery of proteins after gel filtration and affinity purification were 11.6 mg and 1.7 mg, respectively. (**d**) Caseinolytic protease activity of the CV, GF, and AF fractions. One unit of activity is A_280_ unit of change per min. EDTA (4 mM) was included in the reaction to inhibit SVMP. The specific activity of the AF fraction was significantly increased compared to the GF and CV fractions. *p* < 0.0001 (****) compared with the AF fraction. Data are means ± standard deviation; n = 3.

**Figure 3 toxins-15-00500-f003:**
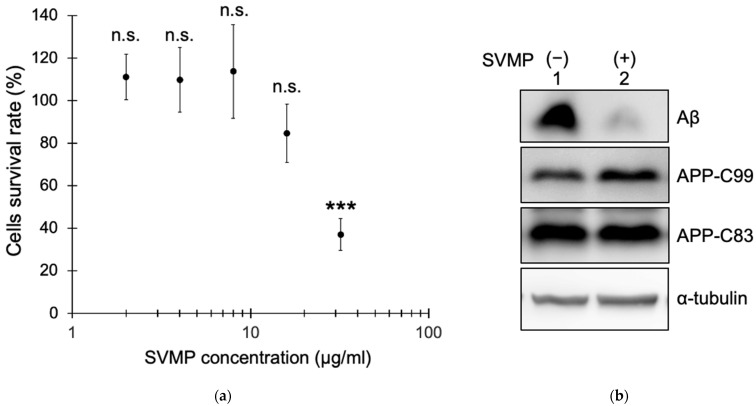
SVMPs reduce Aβ secretion from cells without affecting the levels of APP carboxyl terminal fragments. (**a**) Toxicity of SVMPs to HEK293 cells stably expressing APP Swedish mutant (NLNTK). The cell viability rate was calculated based on the absorbance of water-soluble formazan dye, which is produced upon a reduction in WST-8. Data are means ± standard deviation; n = 3. (**b**) NLNTK cells were treated with 5 μg/mL SVMPs in DMEM for 48 h. Conditioned medium and cells were recovered, and secreted Aβ, cellular APP fragments (C83 and C99), and tubulin were detected by immunoblotting using specific antibodies. (**c**–**e**) Aβ concentration in conditioned medium (**c**) and the intensities of C83 and C99 normalized to that of tubulin (**d**,**e**). Data are means ± standard deviation; n = 3. *p* < 0.001 (***) and *p* > 0.05 (n.s.) compared to the control (no SVMPs).

**Figure 4 toxins-15-00500-f004:**
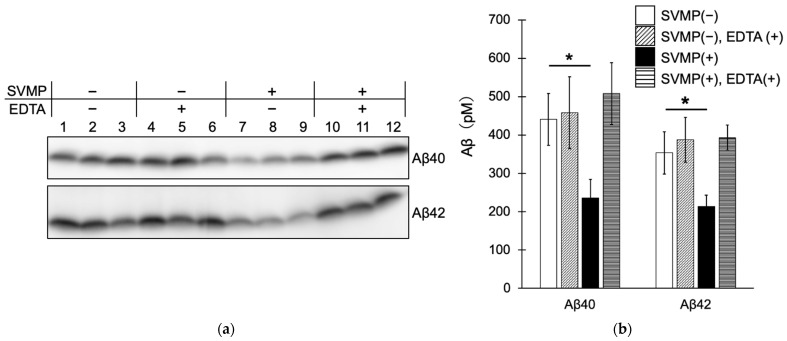
*Protobothrops flavoviridis* SVMPs degrade Aβ40 and Aβ42. (**a**) Aβ40 and Aβ42 were incubated with or without 60 μg/mL SVMPs for 24 h and detected by immunoblotting using an anti-Aβ antibody (82E1). EDTA was used to inhibit SVMPs. (**b**) Aβ concentrations based on immunoblots; means ± standard deviations (n = 3). Aβ concentration was decreased by SVMPs. *p* < 0.05 (*).

**Figure 5 toxins-15-00500-f005:**
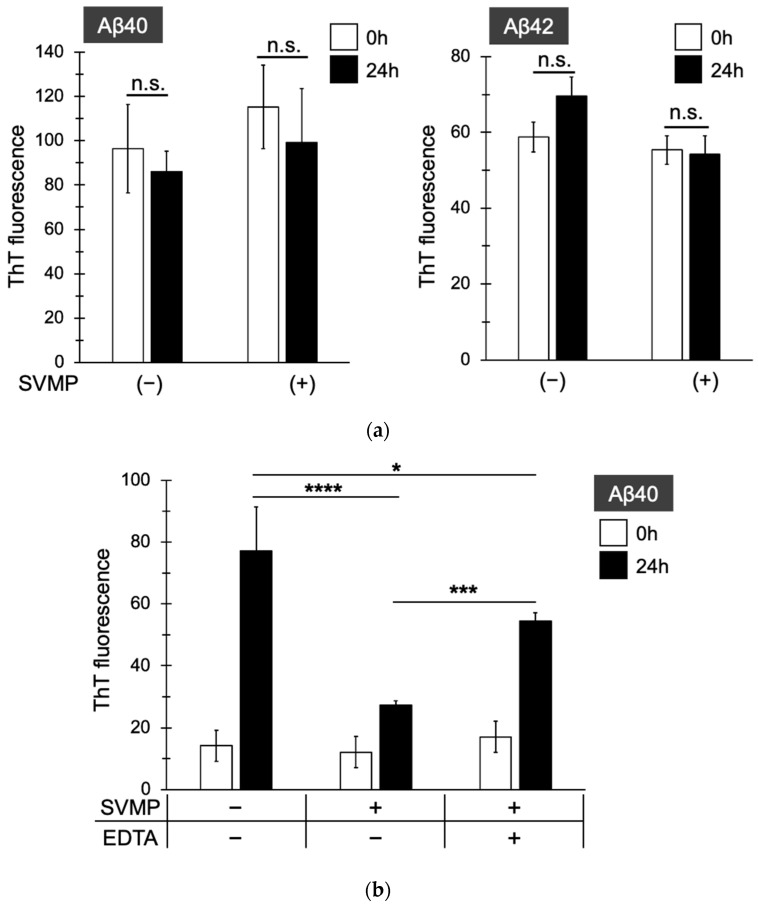
*Protobothrops flavoviridis* SVMPs did not degrade Aβ40 or Aβ42 fibrils but inhibited the formation of Aβ40 fibrils. (**a**) SVMPs were added to pre-formed Aβ40 or Aβ42 fibrils and incubated for 24 h. ThT was added and ThT fluorescence was measured. SVMPs did not decrease the fluorescence of pre-formed Aβ amyloids. (**b**) SVMPs were added to the Aβ fibril-forming reaction and incubated for 24 h. ThT was added and ThT fluorescence was measured. EDTA was added to inhibit SVMPs. Fluorescence intensities are presented as means ± standard deviation (n = 3). *p* < 0.0001 (****), *p* < 0.001 (***), *p* < 0.05 (*) and *p* > 0.05 (n.s.).

**Table 1 toxins-15-00500-t001:** Identification of *Protobothrops flavoviridis* venom metalloproteinases via LC-MS/MS.

Protein ID	SVMP Class	Molecular Mass * (Da)	Length * (Amino Acids)	Sequence Coverage * (%)	Intensity
svMP08_flavoridin	P-II	54,503	483	27.5	5,524,700
svMP06_H2pro	P-I	46,405	408	28.7	4,375,700
svMP09_HR1b	P-III	69,129	615	19.7	1,074,000
svMP07_HR2a	P-II	53,991	480	35.8	903,290
svMP05_HR1a	P-III	68,765	609	6.9	518,200
svMP02_flavorase	P-III	69,015	614	8.0	125,290
svMP03_VMP-III_like	P-III	69,657	620	8.1	70,111
svMP01_HV1	P-III	68,176	612	11.1	37,075
svMP04_jerdonitin_like	P-II	53,872	480	6.5	13,726

* The molecular masses and amino acids length were determined from the sequence information in the database. SVMPs have a signal peptide and propeptide at the amino terminus (1–90), which are cleaved upon maturation. The molecular mass, length, and sequence coverage values encompass these regions.

**Table 2 toxins-15-00500-t002:** Identification of Aβ40 * fragments via LC-MS/MS analysis.

Sequence	Charges	*m*/*z*	Intensity	Ratio
DAEFRHDSGYEVHHQK	3	652.30	320,540	1.0
DAEFRHDSGYEVHHQKLV	4	542.51	14,529	0.045
DAEFRHDSGYEVHHQKLVFFAEDVGSNKGA	5	678.72	694.87	0.0022
IIGLMVGGVV	2	479.29	499.04	0.0016

* Full amino-acid sequence of Aβ40: DAEFRHDSGYEVHHQKLVFFAEDVGSNKGAIIGLMVGGVV.

**Table 3 toxins-15-00500-t003:** Identification of Aβ42 * fragments via LC-MS/MS analysis.

Sequence	Charges	*m*/*z*	Intensity	Ratio
DAEFRHDSGYEVHHQK	3	652.3	428,430	1.0
DAEFRHDSGYEVHHQKLVFFAEDVGSNKGA	5	678.22	98,285	0.23
DAEFRHDSGYEVHHQKLV	4	542.51	4817.5	0.011

* Full sequence of Aβ42: DAEFRHDSGYEVHHQKLVFFAEDVGSNKGAIIGLMVGGVVIA.

## Data Availability

Not applicable.

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
