# Peer review of "A Metalloproteinase Cocktail from the Venom of *Protobothrops flavoviridis* Cleaves Amyloid Beta Peptides at the α-Cleavage Site"

_toxins, 2023, doi:10.3390/toxins15080500_

Round 1

Reviewer 1 Report

REFERENCES

Hartman,G.D.; Egbertson, M.S.; Halczenko, W.; Laswell, W.L.; Duggan, M.E.; Smith, R.L.; Naylor, A.M.; Manno, P.D.; Lynch, R.J.; Zhang, G.; Chang, C.T.C.; Gould, R.J. Non-peptide fibrinogen receptor . Abbreviated Journal Name Year, Volume, page range.

Comment:  Please check ref 16.

In line 83......the acute cerebral infarction drug batroxobin [18] ?

Comment: this sentence is confusing. Please check

In line 355

4.3. Caseinolytic assay A caseinolytic assay was performed [34] to evaluate SVMP activity.....

34. Ono, Y.; Kakinuma, K.; Torii, F.; Irie, A.; Nakagawa, K.; Labeit, S.; Abe, K.; Suzuki, K.; Sorimachi, H. Possible regulation of the conventional calpain system by skeletal muscle-specific calpain, p94/calpain 3. J. Biol. Chem. 2004, 279, 2761-2771.

Comment: Appropriate reference is inside this article [ref 34]. Correct reference is Ishiura et al. Studies of a calcium-activated neutral protease from chicken skeletal muscle. I. Purification and characterization. J Biochem. 1978; 84(1):225-30

24. Kidana, K.; Tatebe, T.; Ito, K.; Hara, N.; Kakita, A.; Saito, T.; Takatori, S.; Ouchi, Y.; Ikeuchi, T.; Makino, M.; Saido, T.C.; Akishita, M.; Iwatsubo, T.; Hori, Y.; Tomita, T. Loss of kallikrein-related peptidase 7 exacerbates amyloid pathology in Alzheimer's dis-ease model mice. EMBO Mol. Med. 2017, 10, e8184.

Comment: Please check the year of publication  

 37. Hori, Y.; Hashimoto, T.; Nomoto, H.; Hyman, B.T.; Iwatsubo, T. Role of apolipoprotein E in β-amyloidogenesis. J. Biol. Chem. 2015, 290, 15163-15174. Role of Apolipoprotein E in β-Amyloidogenesis: ISOFORM-SPECIFIC EFFECTS ON PROTOFIBRIL TO FIBRIL CONVERSION OF Aβ IN VITRO AND BRAIN Aβ DEPOSITION IN VIVO

Comment: Please check TITLE

In figure 3

Comment: Please correct a b c

Reviewer 2 Report

Authors reported the use of an SVMP cocktail derived from the venom of an Asian pit viper, Protobothrops flavoviris, as a potential cleaver of amyloid beta peptides implicated in the pathogenesis of Alzeimer's Disease. The title is appropriate, and authors presented an interesting finding that adds to the discovery of bioactive peptides from venoms as novel therapeutics. The work is well executed and the paper is well written. There are, however, some suggestions for authors with the hope to further improve the manuscript. The comments are given below. 

Abstract: More than half of the abstract is about the introduction of ADAM and SVMP. Only the last 2-3 sentences mentioned about the work done and the finding, in very brief manner. I'd like to suggest authors to simplify the introduction part of the abstract, and include the aim of the study, brief methodology, and concisely include their main findings (results) and discussion with conclusion. 

Line 78-79: Please refine the sentence for clarity: What were these "factors", were they drugs? Pls include any reference, if relevant.

Results: 2.1. Fractionation of SVMP:

Please include information regarding the amount of venom and recovery of SVMP proteins by percentages. These can be put in text, and/or part of the legend/caption of the figure of chromatography. 

Regarding LCMS identification of the SVMP cocktail: Of the many SVMP subtypes, is it possible to include the relative abundance or proportion of each protein in the whole cocktail? This can be added to Table 1. 

Table 1:  Besides the suggested abundance/propertion of the different SVMP proteins, in the caption, pls state how the molecular masses and amino acids lengths were determined. It should be noted these are putative values based on annotation or matching of peptide fragments to database. 

Results: 2.2. , and line 164-165, also Figure 3: Is it possible to include IC50 of the SVMP cocktail against this cell line?

Line 171-173, Figure 3c: On this, I am not sure if there is any need for a positive control (which produced full cleavage of the peptides)?

Line 287-288: In the context of "...to balance cytotoxicity and Abeta-degrading activity." Can author further elaborate on this? What is the significance or relevance of the cytotoxicity / cell viability test, perhaps author can discuss this in a greater extent.

Materials:

Was the venom sourced from one single snake or pooled from several specimens? 

Cell lines - pls include the source. 

The English language is fine and can be understood by general readers in the field.

Reviewer 3 Report

The reviewed article presents the results of a very interesting experiment proving the potential use of SVMPs as therapeutic agents against Alzheimer's disease.

The article is well-written, and the results are correctly presented, however, it is my duty as a reviewer to find areas that could or even should be improved to make the article presentable to a wider readership.

My first comment is to clarify the methodology. Please add what buffer was used to perform SDS-PAGE electrophoresis under non-reducing conditions and what size of the gels were used. Also, please explain why Tris/Tricine electrophoresis was used to analyze the level of secreted AB, while Tris/Glycine electrophoresis was used to analyze C83 and C89. Also, please verify that section 4.8 was definitely about the 50 MDa mass tolerance.

My second comment relates to both the description of the methodology and the results of the experiment on the capacity of casein hydrolysis by purified SVMPs. In section 4.3, it is described that the casein solution was mixed with the SVMP fraction as if it were the AF fraction. In contrast, the results presented in Figure 2 show that G2 and crude venom were also analyzed. There is also a lack of information on whether the three samples used in the experiment were standardized in terms of protein concentration so that they could be directly compared as was done in Section 2.1.

Reviewer 4 Report

Review of the article: A metalloproteinase cocktail from the venom of Protobothrops flavoviridis cleaves amyloid beta peptides at α-cleavage sites

Manuscript ID: toxins-2543619 

In this study, the authors hypothesized that SVMPs may have therapeutic potential for AD and demonstrated that pooled SVMPs from Protobothrops flavoviridis venom reduced Aβ production by cultured human cells and degraded synthetic Aβ40 and Aβ42 peptides at α-site of APP; SVMPs did not degrade Aβ fibrils, but interfered with their formation. The information in this study would be beneficial to find drugs with better therapeutic potential. I have few comments that the authors should notice and reply, as bellows.

1.     Before conducting the parametric statistics such as one-way analysis of variance, the normality and homoscedasticity of data need be to tested first and the results of such tests should be revealed to testify they are not violated.

2.     Some paragraphs in Introduction and Discussion are not properly expressed and need to be revised. Related to the penultimate paragraph of Introduction, the authors should also summarize the drugs that have already been tested or used in AD therapy and correlate them to the therapeutic potential in AD therapy by using snake venoms. In specific paragraph(s) of Discussion, the authors should focus on discussing the strengths and weaknesses of using SVMPs in AD therapy. For example, Ref. 28 demonstrated that matrix metalloproteinase-9 degrades amyloid-β fibrils in vitro and compact plaques in situ. Does it mean matrix metalloproteinase-9 has better therapeutic potential than the SVMPs in your study? In addition, the paragraph in lines 253-271 is not good to read and need to revise.

3.     At Fig 2(c), the labels “M CV1 GF2 AF3” are slightly shifted to left.

4.     At line 145, “A2 fraction”?

5.     At line 157, “molecular weight” or “molecular mass”?

6.     At lines 161-162, 176-177, the two “cells” in the sentence are confusing.

7.     At Table 2 (1), the ratio “0.0045” is wrong.

8.     At lines 225 and 227, “Tht” should be “ThT”?

9.     At line 381, “Section 5.5” should be “Section 4.5”.

10.  At line 484, some information of this reference is missed.
